# From Satirical Poems and Invisible Poisons to Radical Surgery and Organized Cervical Cancer Screening—A Historical Outline of Cervical Carcinoma and Its Relation to HPV Infection

**DOI:** 10.3390/life14030307

**Published:** 2024-02-27

**Authors:** Leonard Jung, Gilbert Georg Klamminger, Bert Bier, Elke Eltze

**Affiliations:** 1Institute of Pathology, Saarbrücken-Rastpfuhl, 66113 Saarbrücken, Germany; 2Department of General and Special Pathology, Saarland University (USAAR), 66424 Homburg, Germany; 3Department of General and Special Pathology, Saarland University Medical Center (UKS), 66424 Homburg, Germany

**Keywords:** cervical cancer, HPV, cervical cancer screening

## Abstract

Over the last century, the narrative of cervical cancer history has become intricately tied to virus research, particularly the human papillomavirus (HPV) since the 1970s. The unequivocal proof of HPV’s causal role in cervical cancer has placed its detection at the heart of early screening programs across numerous countries. From a historical perspective, sexually transmitted genital warts have been already documented in ancient Latin literature; the remarkable symptoms and clinical descriptions of progressed cervical cancer can be traced back to Hippocrates and classical Greece. However, in the new era of medicine, it was not until the diagnostic–pathological accomplishments of Aurel Babeş and George Nicolas Papanicolaou, as well as the surgical accomplishments of Ernst Wertheim and Joe Vincent Meigs, that the prognosis and prevention of cervical carcinoma were significantly improved. Future developments will likely include extended primary prevention efforts consisting of better global access to vaccination programs as well as adapted methods for screening for precursor lesions, like the use of self-sampling HPV-tests. Furthermore, they may also advantageously involve additional novel diagnostic methods that could allow for both an unbiased approach to tissue diagnostics and the use of artificial-intelligence-based tools to support decision making.

## 1. Introduction

From an epidemiological perspective, cervical cancer is a global disease with pronounced regional variations. Being the fourth most common neoplasm and, therefore, the second most common cause of cancer-related deaths globally [1,2], incidences vary drastically per region, affecting public health systems predominantly in developing countries. Although in Eswatini, incidence rates of up to 95.9/100,000 per year have been reported [3], European incidence rates are 10.6/100,000 per year [3], and an even lower incidence rate has been reported in Palestine (2/100,000 per year). In Germany, it is ‘solely’ the 14th most frequent tumor in women and, therefore, only the fourth most common gynecological malignancy [4,5]. Its political dimension is extremely diverse in itself: in a way, it reflects differences between social standards (The socioeconomic factor is a major risk factor for the development of cervical cancer [6]) as well as health and economic issues relating to adequate screening services. With the overall aim of mortality reduction, organized screening programs have proven to reduce incidence rates as well as cervical-cancer-dependent mortality rates in Northern Europe. To this day, the clear advantage of organized screening programs is widely evident [4,7,8,9,10].

The medical–therapeutic dimension is subject to constant change; nowadays, diagnostic dogmas of cytology are increasingly replaced by primary gene amplification testing. Moreover, therapeutic strategies are constantly being improved [11,12,13]. In the present review, we touch on all these aspects merely, thereby supplementing the body of known literature on this disease with a historical perspective. Despite it being worth reading reviews of the historical development of cervical cancer screening, our report aims to honor achieved surgical, pathological, virological, and potential future milestones in equal measure, providing a comprehensive overview for gynecologists, pathologists, and scientists involved in cervical cancer research.

## 2. Materials and Methods

A literature search was conducted via the online databases PubMed and Google Scholar. Using the Boolean AND/OR operators, we combined the tags “history”, “cervical cancer”, “cervical neoplasm”, “HPV”, “oncogenic virus”, “Harald zur Hausen”, “Pap-smear”, “HPV-test”, and “cervical cancer screening”, along with associated mesh terms. Affiliated literature as well as references of semantically consistent articles were considered and included at our convenience. After the initial title/abstract screening, relevant papers were discussed and individually selected upon the agreement of all the authors. Literary works written in languages other than English, German, French, Italian, and Latin, as well as solely brief historical summaries, were excluded.

## 3. From Ancient Promiscuity to Microscopic Diagnosis

The first description of cervical cancer and its symptomatic irregular bleedings was written 400 years BC by Hippocrates, who, recognizing the fatal course of the disease, prescribed a palliative concept [14]. However, the phylogenetic development of papillomavirus genomes and, thus, the main risk factor for the development of cervical cancer (id est infection with HPV), can be traced back several hundred million years; as a result, an affection of women since early human history seems quite plausible [15]. Early written evidence proves that sexually transmitted diseases have been recognized already in ancient Rome, where genital warts were interpreted as a sign of promiscuity. The warts were called *ficus*, corresponding to their fig-like appearance. The suspected connection between anal warts and anal intercourse is revealed by satirical poems of that time [16]:

“*Ut pueros emeret Labienus vendidit hortos.*

*Nil nisi ficetum nunc Labienus habet*.”

“In order to buy boys, Labienus sold his gardens,

but now Labienus only possesses a fig tree.”

(M. Valerii Martialis Epigrammaton Libri XII, XXXIII)

“*Ficosa est uxor, ficosus et ipse maritus,*


*Filia ficosa est et gener atque nepos,*



*Nec dispensator nec vilicus ulcere turpi*



*Nec rigidus fossor, sed nec arator eget.*



*Cum sint ficosi pariter iuvenesque senesque,*



*Res mira est, ficos non habet unus ager.”*


“The wife is full of genital warts; even the husband is full of genital warts.

The daughter is full of genital warts as well as the son-in-law and the grandson.

Neither the steward nor the farm bailiff is free from the distasteful ulcer;

nor even the harsh digger or the ploughman is without.

Since both the young and the old are full of genital warts,

the incident is astonishing; the acre does not have figs.”

(M. Valerii Martialis Epigrammaton Libri VII, LXXI)

It was not until the beginning of the 19th century and the rise of modern gynecology, that the epidemiologic risk factors as well as the diagnostic and treatment capabilities of cervical cancer improved. As a key factor, the expansive use of the vaginal speculum, supported by the French gynecologist Joseph Récamier between 1801 and 1819, provided deeper clinical and anatomical insights into the internal female reproductive organs and the associated possibility for initiating further diagnostic and therapeutic measures [17]. Historically, finds of ancient specula in the ruins of Pompeii prove the longstanding tradition of this particular examination tool, even though up to this time, it was largely associated with prostitution as well as venereal diseases and was, therefore, socially stigmatized [15,18]. Following the successful removal of the cervix by Nikolaas Tulp in the 17th century [15], Friedrich Benjamin Osiander from Göttingen amputated a cancerous cervix in 1801 [19]. In 1813, Bernhard von Langenbeck successfully performed a vaginal hysterectomy in a patient with cervical carcinoma [20].

Apart from these therapeutic successes, the Italian doctor Domenico Rigoni-Stern was one of the first to study the epidemiology of cervical cancer by analyzing the death certificates of women who died between 1760 and 1839 in 1842. He thereby recognized that cervical cancer was more common in widows and prostitutes—whereas virgins and nuns were rarely affected—and drew the conclusion that cancer of this type must be associated with sexual contact [21]. The German pathologist Carl Arnold Ruge and the German gynecologist Johann Veit laid the foundation for further systematic research into cervical carcinoma by investigating and describing gynecological surgical specimens removed by the gynecologist Karl Schröder (He suggested high cervical resection as a treatment option at the time.) [15]; in 1877, they described cervical carcinoma as an independent disease and highlighted the impact of presurgical biopsies [15,22].

Though surgical therapies at the time were chosen intuitively, the advantage of uterus removal in terms of overall survival was pointed out consecutively. Early works, such as “*Uterus Amputation am vaginalen Wege*” (1830) and “*Ca colli uteri*” (1878, *Wiener medizinische Wochenschrift*), focused on (radical) vaginal uterus removal [23]. The first radical abdominal hysterectomies were performed in 1878 by the German gynecologist Wilhelm Alexander Freund and in 1880 in Prague by Karel Pawlik [24,25]; subsequently, this procedure of a ‘radical hysterectomy’ was summarized in 1898 by the Austrian gynecologist Ernst Wertheim, who assumed a spread of neoplastic cells into the lymphatic tissue and, therefore, added a systematic pelvic curative lymphadenectomy to the surgical protocol, using individualized forceps made for dissecting the dorsal parametria (“*Wertheim Klemme*”). This modification would be, later on, further developed by the American Joe Vincent Meigs [26]. Such radical approaches were accompanied by an improvement in the long-term prognosis compared to the vaginal hysterectomy introduced by Frederich Schauta, whereby the latter had the advantage of a lower perioperative mortality rate in the 19th century [15]. An ongoing dispute between Schauta and Wertheim (so-called “*Drüsenstreit*”, the nodule dispute) regarding the importance of lymphadenectomy filled the professional European literature for decades and was not resolved until Wertheim’s death and the introduction of radiation as a method of treatment [27].

In the late twentieth century, Daniel Dargent and Marc Possover proposed a combined laparoscopic lymphadenectomy with vaginal specimen removal, including a nerve sparing technique [28,29], within the overall concept of TRLH (total radical laparoscopic hysterectomy). Furthermore, Michael Hoeckel and Rainer Kimmig suggested the TMMR (total mesometrial resection) method as a potential treatment option [30,31]. Within the last century, radical hysterectomies have been divided into distinct subtypes based on the extent of the resection of the parametria/paracervix, and systematic pelvic lymphadenectomies have mostly been replaced by the concept of the sentinel lymph node mapping (SLNM). Recently, the LACC (Laparoscopic Approach to Cervical Cancer) trial provided a paradigm shift by demonstrating the advantage of open abdominal surgery in cervical cancer patients over laparoscopic tumor resection [32].

To date, surgical therapy (abdominal open radical hysterectomy combined with SLNM or/and systematic pelvic lymphadenectomy) is considered as a first-line treatment of locally advanced cervical cancer, although definitive radiotherapy with concomitant chemotherapy is favored in the progressed disease. Current treatment strategies avoid the combination of radical surgery and radiotherapy owing to its high morbidity rate [6]. To reduce postoperative morbidity and complications, experimental setups nowadays question the need for radical surgical therapy in patients with 1A2 or 1B1 low-risk cervical carcinomas with lesions smaller than or equal to the size of 2 cm [33].

Another diagnostic milestone in the early detection of cervical carcinoma was the invention of colposcopy by the German gynecologist Hans Hinselmann in 1925 [34]. Despite his involvement in Nazi Germany and the resulting slowdown in the spread of his medical achievement, he was undoubtedly an expert in the field of the clinical detection of cervical neoplasia; he was flown to Argentina as a consultant shortly before Eva Perón’s death [35]. Two years later, a differentiation between normal cervical mucosa and pathologically altered mucosa based on an iodine reaction (Lugol’s solution) was achieved by the Austrian gynecologist and pathologist Walter Schiller [36]. Following up the discovery of the British doctor Hayle Walsh in the middle of the 18th century that a distinct identification of cancerous cells was possible using solely a light microscope [37,38], the Romanian doctor Aurel Babeş examined cells from a female cervix in 1927, which he obtained using a platinum loop, under a microscope. Thereby, he succeeded in detecting the presence of cervical cancer [38]. At the same time, the Greek physician George Nicolas Papanicolaou was conducting research in America after completing his doctorate in Germany (Munich) [39]. He transferred his research findings on menstruation-related cytological changes in the vaginal mucosa of guinea pigs to humans and discovered—with the active support of his wife as a repeated volunteer—the same cyclical cellular estrogen dependency and, by chance, neoplastic cell forms [40]. In 1928, he presented his work for the first time [41,42], which, after technical improvements by the Canadian gynecologist J. Ernest Ayre, provides an efficient tool for the early detection of cervical cancer as it is used today [43].

See Figure 1 for an overview of historical diagnostic and surgical achievements and Figure 2 for an assessment of HPV infection during today’s routine cervical cancer screening via light microscopy.

## 4. The Presence of Oncogenic Viruses: From “Invisible Poison” to Viral Genetics

Although scientifically supported by epidemiological studies showing higher incidences in women who (themselves or their husbands) had a high number of sexual partners (*n* > 2) [44,45] or were regularly involved in prostitution [46,47], all the attempts linking cervical cancer to sexually transmitted infections remained unsuccessful within the second half of the 19th century. Strikingly, the prevalence was lower among Jewish women and in women married to circumcised males [48]. Viruses came into question as a possible sexually transmitted pathogen, whereas at that time, the history and study of viruses was a comparatively young research field. In 1894, Dmitri Ivanovsky succeeded in transmitting the mosaic disease of tobacco from infected to healthy tobacco plants by pressing leaves infected with the mosaic disease of tobacco through a filter that only allowed a pass through for pathogens smaller than bacteria. As the final cause of the disease, he suspected an “invisible poison” [49]. In accordance, Martinus Beijerinck achieved the same results in 1898, and he called the pathogenic agent a *virus* (*poison/sap/mucus*). Furthermore, Beijerinck defined two further characteristics of a virus, namely, the ability to pass through a filter that only allowed a pass through for particles smaller than bacteria and the necessary presence of living cells to grow [50]. The proof that this new infectious agent may also infect humans and thereby even cause disease was obtained by Walter Reed in 1902. The yellow fever virus is considered as the first proven human pathogenic virus [51], and the acceptance of the scientific world that infectious diseases in humans could be caused by viruses as well as bacteria provided the basis for upcoming research tasks in the field of virology. In addition to research of viral infectious diseases, the potential involvement of viruses in the development of neoplasia was investigated already at an early stage. In 1911, Peyton Rous succeeded in triggering a tumor in chickens with the help of a virus. Three years earlier, two Danish researchers, namely, Vilhelm Ellerman and Oluf Bang, were able to provoke leukemia in chickens using a cell-free extract, which served as the first evidence suggesting a link between viruses and tumors [52]. Peyton Rous once more and J.W. Beard were successful in 1936 in transferring the same warts from infected to healthy areas of skin by subcutaneous injection using a cell-free extract of existing lesions. Their efforts were based on observations made by Richard E. Shope, who found warts on the heads of American wild rabbits and concluded that they were caused by a wart virus. As Beard and Rous described, they first recognized benign tissue growths, which appeared at the injection site. These irregularities then turned malignant, spread, and finally killed the rabbit [53]. In 1961, Yohei Ito and Alfred Evans demonstrated the role of genetic viral material as a decisive factor in the development of skin cancer in rabbits [54]. Up to this point, research into links between neoplasia and viruses had been limited to animal studies. Finally, Michael Anthony Epstein, Yvonne Barr, and Bert Achong succeeded in detecting the Epstein–Barr virus in Burkitt’s lymphoma in 1964 [55] and linked neoplasia in humans and a human pathogenic virus for the first time. That said, a final link between human papillomaviruses and cervical cancer was inconceivable until the research findings of Harald zur Hausen. In 1965, June Almeida proposed the existence of different human wart viruses [56], and in 1968, Rawls et al. (among others) emphasized herpes simplex type 2 as a possible causative agent of cervical cancer [57,58]—an assertion that was not only disproved by zur Hausen’s abortive attempts to isolate genetic material from herpes simplex viruses in cervical tumors [59] but also refuted by a prospective study in the mid-1980s [60]. Alternatively, zur Hausen et al. began to hypothesize a conditional relation between cervical cancer and human papillomaviruses in the early 1970s, based on reports in the medical literature of the rare, malignant transformation from genital warts (*Condyloma acuminata*) to squamous cell carcinomas [59,61,62]. HPVs were finally detected in cervical carcinomas in 1983 [63,64]. Approximately 50% of the tissues that were examined contained the genetic material of HPV 16, whereas in 1984, HPV type 18 was detected in 20% of the examined tumors [65,66]; subsequently, papillomaviruses 16 and 18 were categorized as high-risk variants. Consecutive studies provided increasingly precise and incontrovertible evidence of the role of high-risk HPVs in the development of cervical carcinoma: This included, among others, the decisive discovery that the genetic material of papillomaviruses can be incorporated into the host cell’s genome of cancer cells and that specific viral genes (coding oncoproteins E6 and E7) thereby switch off antiproliferative cell mechanisms [67,68]. In 1991, McDougall et al. demonstrated that immortalized human keratinocytes spontaneously developed malignant degeneration after long-term cultivation in the laboratory [69]. Additional epidemiological evidence of an HPV association with cervical carcinoma was provided in the late 1980s and early 1990s [70,71]. As a result of this growing body of evidence, the International Agency for Research on Cancer classified HPVs 16 and 18 as carcinogenic, HPVs 31 and 33 as probably carcinogenic, and other HPV types as possibly carcinogenic agents in 1995 [72]. The involvement of HPVs in cervical cancer, viz., the group of HPV-positive tumors, changed over the course of time as further HPV types were characterized. The proportion of HPV-positive tumors of the cervix uteri was estimated at 72% at the beginning of the 1980s [63] and was raised to 99.7% in 1999 [73]. Consequently, researchers were starting to call for a rethink of cytology as a routine examination method in cervical carcinoma screening: The idea for a combination of cytology and HPV testing or single HPV testing was born [73]. Up to now, over 200 genetically differentiated HPV types have been detected [74] (of which 40 can infect the genital tract via the mucosal epithelium, and 12 virus types are categorized as high-risk variants in the development of cervical cancer [75,76]), although solely 60 different HPV types were identified toward the end of the 1980s [77]. In 2008, Harald zur Hausen (*11.03.1936; † 29.05.2023) was honored with the Nobel Prize in physiology and medicine for his most valuable research [78]. Table 1 summarizes the HPV types associated with cervical cancer and their capabilities to infect the genital area.

The established role of the human papillomavirus in cervical cancer has sparked the interest of the pharmaceutical industry to develop a vaccine against HPV. Before Gardasil^®^, a recombinant, quadrivalent vaccine against HPVs 6, 11, 16, and 18, was approved by the FDA (Food And Drug Administration, USA) on 8 June 2006 [79] and Cervarix^®^, a recombinant vaccine against HPV high-risk types 16 and 18, was authorized by the European Medicines Agency in 2007 [80], as well as Gardasil^®^-9, the successor of Gardasil^®^, which, in addition to the aforementioned virus types, also covers virus variants 31, 33, 45, 52, and 58, was finally approved by the FDA in December 2014 [81], the foundations for the successful introduction of the vaccine were laid back in the 1990s, when Kirnbauer and associates used papillomavirus clones to successfully produce HPV 16 virus-like particles and attested to their strong immune response [82]. By 2023, the pharmaceutical company Merck & Co., Inc. (Rahway, NJ, USA), which sells Gardasil, is expected to generate sales of up to $2.5 billion—and the forecasts for the next years are still rising [83]. Cecolin^®^ (Xiamen Innovax Biotech Co., Ltd., Xiamen, China)—another vaccine that was approved in China in 2020—is a bivalent vaccine and, thus, covers HPV types 16 and 18. Because its immunogenicity and properties are yet to be investigated, approval by US and European authorities are uncertain so far [84]; nevertheless, it was already prequalified by the WHO in 2021 [85]. In the near future, an Indian vaccine should be available, after its permission by the Indian administration [76]. To date, 120 million women have been vaccinated with a minimum of one dose, mainly in high-income countries [86]. As new vaccines aim to show lower costs [87], broader coverage of administered vaccinations in low-income countries could strengthen the WHO’s goal for eliminating cervical cancer [88]. However, to date, official screening recommendations do include all women, regardless of their vaccination status, because HPV vaccination is not able to provide sufficient protection against all the cancerogenous HPV virus types [4]. For a visualization of the history of HPV research in cervical cancer, refer to Figure 3.

## 5. The HPV Test and Its Role in Cancer Screening

The first efforts to establish a commercial HPV test took place just 18 months after the eminent role of HPVs in the pathophysiology of cervical carcinomas gained attention [89]. The first steps in the development of a commercial HPV test can be traced back to the USA, with the first official FDA approval recorded in 1988 (FDA premarket approval P880009). However, the ViraPap^®^ kit from BRL–Life Technologies (Bethesda Research Laboratories–Life Technologies, Bethesda, MD, USA), in cooperation with Georgetown University (Washington, DC, USA), did not achieve a clinical breakthrough because there were insufficient data that could demonstrate a direct patient benefit [90]. As the test was radioactive, it had a limited shelf life and was potentially dangerous to the staff using it. Subsequently, Digene’s (Gaithersburg, MD, USA) Hybrid Capture^®^ 2 high-risk HPV DNA test was officially approved by the FDA in 1999 as a triage test for atypical squamous epithelial cells of undetermined significance (ASC-US) [91]. In 2001, the American Society for Colposcopy and Cervical Pathology published the consensus guidelines for the management of women with cervical cytological abnormalities, in which Digene’s Hybrid Capture^®^ 2 high-risk HPV DNA test was proposed as a secondary test for atypical squamous epithelial cells of uncertain significance (ASC-US) [92]. This recommendation was based on the advantages of the HPV test, filling the missing space between a possible overtreatment as well as missed neoplasms owing to non-compliance with repetitive cytological testing follow-ups in the case of a single suspicious cytological-screening result. Therefore, the American Society for Colposcopy and Cervical Pathology recommended an HPV test after a suspicious cytological result before women should receive a colposcopic examination. This approach was not without controversy. The *ASCUS/LISL Triage Study for Cervical Cancer* (ATLS) [93], the research of which was used to develop guidelines for the American Society for Colposcopy and Cervical Pathology and which confirmed that the HPV test had a sensitivity of 96.3% in detecting CIN III, was called into question fairly quickly after its publication [93]—opponents complained that routine clinical use did not provide certainty in the detection of cellular atypia [94,95]. In 2002, the American Cancer Society recommended HPV testing using the Digene HC2 high-risk HPV DNA test (QIAGEN, Venlo, The Netherlands) as an additional test option, namely, as a primary co-test every three years, for women over the age of 30. This was confirmed by the FDA in 2003 and was included in the American College of Obstetricians and Gynecologists’ practice bulletin [91,96,97,98]. Opposite to that, the United States Preventive Services Task Force (USPSTF) interpreted the scientific data and the derived value of HPV testing as being insufficient and did not recommend its routine use as a primary tool in the screening program [99]. To counter stagnating economic growth in 2005, Digene penned the concise slogans “You’re not failing your Pap test, but it might be failing you.” and “If you’re a gambling woman, then getting just a Pap test is fine.” [100] with success! According a consumer survey, this precautionary advertising led to a significant increase in interest in HPV testing and the cervical-cancer-screening program—within a six-month observation period following the advertisement, a 42% increase in revenue for the HPV test was recorded [101]. Owing to the high negative predictive value of HPV testing, the USPSTF, the American College of Gynecologists and Obstetricians, and the American Cancer Society recommended co-testing with HPV testing and cytology at five-year intervals as a primary screening algorithm in 2005, overcoming past contradictions [102,103,104]. In 2014, the FDA approved the HPV test COBAS from Roche (Basel, Switzerland) as a primary screening tool for cervical cancer screening in women over 25 [105].

To date, numerous commercially available HPV-testing systems have received approval by the FDA and EMA. The presence of viral infections can be determined either by means of direct genome detection or DNA/RNA amplification. Related to the mode of the HPV detection as well as the specific test system, different HPV types (low risk/high risk) are simultaneously screened. Even though polymerase chain reaction (PCR)-based methods allow for a high degree of sensitivity, the clinical ability of single testing in detecting high-grade squamous lesions remains to be resolved [106]. In comparison, the predictive value of hybrid-based methods is inherently correlated with the quality and quantity of the specimens [107]. Finally, the selection of individual test systems should furthermore be made in accordance with regional approvals by federal agencies as well as the capabilities and infrastructures of hospitals’ local screening programs.

The integration of HPV testing into the cervical-cancer-screening program took a little longer in Germany. Introduced in 1971, the German cervical-cancer-screening program consisted initially of annual pap tests. The first attempts to establish HPV testing within the German screening program were made in 2006 but failed at such an early time. Regulated by the framework conditions of the German legal system and, in particular, ‘*Sozialgesetzbuch V*’, every upcoming change in cancer screening is associated with a change in the German law. Potential modifications to early-cancer-screening programs must be proposed by the Federal Joint Committee to be brought to a binding legal form by the legislature. The Institute for Quality and Efficiency in Health Care (IQWiG), which is responsible for submitting scientific assessments and scientific methods to the Federal Joint Committee, was commissioned in spring 2010 to assess the benefits of the HPV test in primary screening for cervical carcinoma; the report was published in 2012 and postulated a benefit for HPV testing with regard to reductions in high-grade cervical intraepithelial neoplasia and the incidence of cervical carcinoma [108]. In response, the Federal Joint Committee commissioned IQWIG to develop a letter of invitation for screening programs and an information package on cervical cancer that women aged from 20 to 60 should receive from their health insurers. Additionally, women > 30 years should receive an HPV test every five years, and cytology should be reserved as a triage, with the overall aim to collect data during a six-year transition phase, which could then ultimately be analyzed to establish a superior screening strategy [109]. The early detection program was adjusted again on 22 November 2018: Women > 35 years should be offered a co-test consisting of a pap smear and HPV testing at three-year intervals; the annual single cytology test remained as a part of the cancer-screening program for women aged between 20 and 34 [110]. Following the national German guideline on the prevention of cervical cancer, the cancer-testing systems used in clinical routines should be able to identify high-risk HPV types and should cover a sensitivity of at least 95% (The specificity should be 98% at a minimum) in detecting cervical intraepithelial neoplasia grade II (CIN II, high-grade squamous intraepithelial lesions). Nationwide, commonly implemented test systems include, for example, the BD Onclarity™ HPV test (Becton Dickinson, NJ, USA) or the PapilloCheck^®^ HPV test (Greiner Bio-One, Kremsmünster, Austria) [4]. Table 2 offers a synopsis of the approved and commercially available HPV tests that meet the criteria of the German guideline to prevent cervical cancer.

According to the ‘WHO Guideline for the Screening and Treatment of Cervical Pre-Cancer Lesions for Cervical Cancer Prevention’, only a globally applying screening algorithm favoring HPV testing has been established to date [111]. Still, largely different recommendations across American and European guidelines exist, considering not only regional differences in resources, such as laboratory landscapes and testing capabilities, but also the available access to health systems among different socioeconomic groups, even in high-income countries and, therefore, diverging proposed starting ages and HPV test/pap smear intervals within individual screening programs [112,113,114,115,116]. Besides the modern approach for the dual staining of cytological specimens (a p16/Ki67 staining marks HPV association as well as cell proliferation [117]), a meta-analysis carried out by Ogilvie et al. advantageously shows that an HPV test can also be carried out independently by the patient (self-collected vaginal specimens) in self-sampling, with comparable test reliability, rendering the possibility to potentially better include access-limited populations as well as patients with trauma history within screening programs [118].

## 6. The Future of Cervical Cancer Screening—Quo Vadimus?

Despite the urge for quick offers and straightforward access to HPV vaccination (primary prevention), the benefits of an unbiased approach to tissue diagnosis will remain the key in upcoming diagnostic developments. Therefore, novel screening techniques as well as methods for analysis could find their way from research uses to clinical applications in the near future. Because vibrational spectroscopy generates an individual molecular spectroscopic signature of tissues, it provides a quick, inexpensive, and non-destructive way for observer-independent tissue analysis [119]. It has been already successfully employed at a single-cell level to differentiate between normal cells and cells deriving from cervical intraepithelial neoplasia, using Raman spectroscopy on regular cervical cytology samples [120] as well as on tissue samples to distinguish between adenocarcinoma and squamous cell carcinoma [121] or among cervicitis, precursor lesions, and invasive carcinoma [122]. Additionally, the future employment of AI (artificial intelligence) applications and machine learning may lead to objective, independent, and simple diagnostic methods and, therefore, expand the fields of the screening and early diagnosis of cervical cancer [123]. Deep-learning software and suitable algorithms that evaluate images from the cervix could, at a certain point, outperform the current diagnostic ability in terms of the accuracy and reproducibility of colposcopy-guided diagnosis [124,125]. Furthermore, it could reduce costs by lowering the need for laboratory equipment and highly skilled staff [126]. In progressed disease, AI may be capable to improve the workflow of MRI image segmentation [127] and may noninvasively be used to identify lymph node metastasis in cervical cancer [128]. Another potential future improvement could make use of the prognostic impact of DNA methylation patterns. Epigenetic structures may, therefore, provide a new approach in diagnostics and individualized medicine to distinguish the risk patterns of the transformation. Hypermethylated genes seen in precancerous lesions could possibly be taken into account when it comes to the triage of women with a higher risk for developing cancer [129,130]. In the next step, this could lead to new pharmacological therapies in cervical neoplasia: Trichosanthin, the main substance in a Chinese medical herb, for instance, is able to reactivate tumor suppressor genes by inhibiting DNA methyltransferase in cervical cancer cells [131]. To date, modern treatment options include immunotherapeutic drugs such as pembrolizumab, which is yet to be approved for cervical cancer patients [132]. In addition, ongoing research trials examine inter alia the combination of immunotherapeutic and vaccination approaches [133] as well as the combination of checkpoint inhibitors with radiation therapy [134], providing hope for patients with, so far, limited treatment options.

As research continues, both the early detection as well as therapeutic options of cervical cancer could be revolutionized; however, the future path of the development can only be critically evaluated retrospectively. The WHO’s ambitious future goal is clearly defined: to eliminate the worldwide burden of cervical cancer by 2030 through the vigorous implementation of primary, secondary, and tertiary prevention strategies [88].

## Figures and Tables

**Figure 1 life-14-00307-f001:**
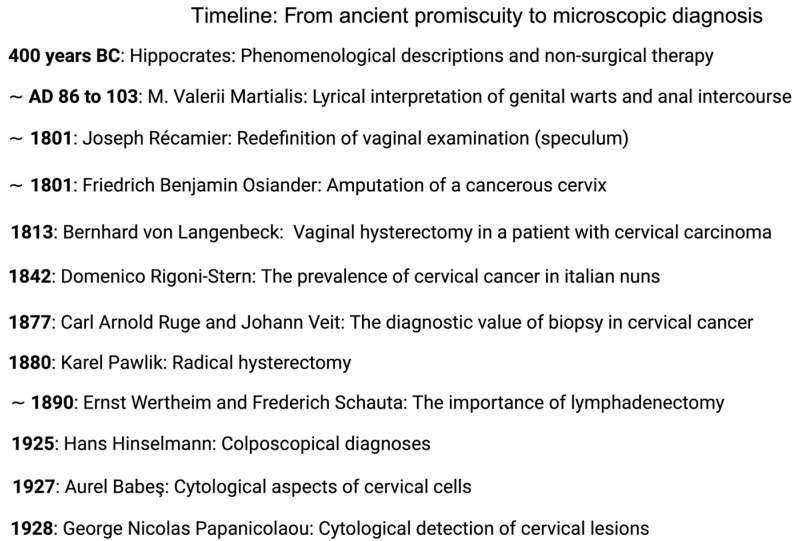
Overview of diagnostic and surgical achievements from Hippocrates to Papanicolaou.

**Figure 2 life-14-00307-f002:**
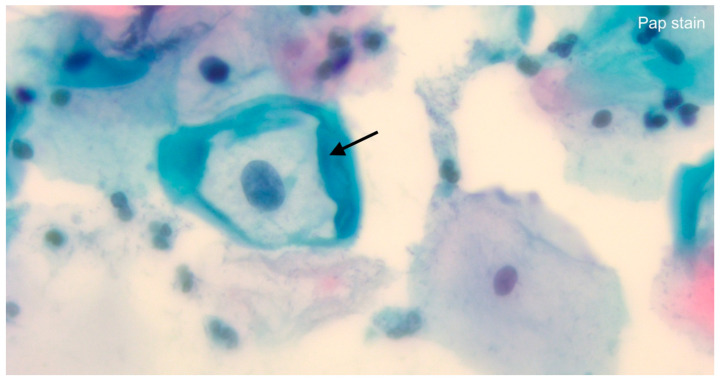
Distinct cytopathological features (slightly enlarged acentric nucleus, perinuclear cytoplasmatic vacuolization, surrounding prominent cytoplasm) mark koilocytotic atypia, an HPV-related transformation from squamous cells to ‘koilocytes’ (black arrow); lens magnification: ×63.

**Figure 3 life-14-00307-f003:**
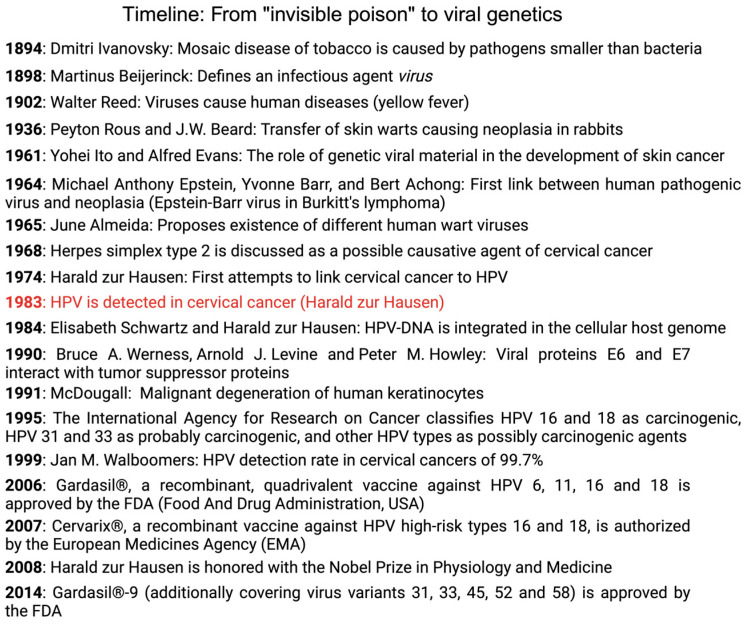
Overview of historical virological milestones from Beijerinck to zur Hausen.

**Table 1 life-14-00307-t001:** HPV types and their classification according their malignancy potential by the IARC.

HPV Types	IARC Classification (2012)
16; 18; 31; 33; 35; 39; 45; 51; 52; 56; 58; 59	1 (carcinogenic; high risk)
68	2a (probably carcinogenic)
26; 30; 34; 53; 66; 67; 69; 70; 73; 82; 85; 97	2b (possibly carcinogenic)
Gamma-HPV; Beta-HPV; 6; 11	3 (not classifiable)
40; 42; 43; 44; 54; 61; 72; 81	low-risk types

**Table 2 life-14-00307-t002:** Commercially available HPV tests, which meet the criteria of the German guideline to prevent cervical cancer.

HPV Test	HPV Types	Technique
Digene Hybrid Capture 2 High-Risk HPV DNA Test (QIAGEN, Gaithersburg, Inc.)	16; 18; 31; 33; 35; 39; 45; 51; 52; 56; 58; 59; 68	DNA: whole genome
COBAS HPV Test (Roche Diagnostics)	16; 18; 31; 33;35; 39; 45; 51; 52; 56; 58; 59; 66; 68	DNA: L1
Cervista™ HPV HR and GenFind™ DNA Extraction Kit (Hologic)	16; 18; 31; 33; 35; 39; 45; 51; 52; 56; 58; 59; 66; 68	DNA E6/E7/L1
APTIMA HPV Assay (Hologic)	16; 18; 31; 33; 35; 39; 45; 51; 52; 56; 58; 59; 66; 68	RNA E6/E7
Abbot RT High-Risk HPV Test	16; 18; 31; 33;35; 39; 45; 51; 52; 56; 58; 59; 66; 68	DNA L1
BD Onclarity HPV Test	16; 18; 31; 45; 51; 52; 33–58; 56–59–66; 35; 39–68	DNA E6/E7
PapilloCheck^®^ HPV Test	16; 18; 31; 33; 35; 39; 45; 51; 52; 53; 56; 58; 59; 66; 68; 70; 73; 82; 6; 11; 40; 42; 43; 44	DNA E1

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
