# Peer review of "From Satirical Poems and Invisible Poisons to Radical Surgery and Organized Cervical Cancer Screening—A Historical Outline of Cervical Carcinoma and Its Relation to HPV Infection"

_life, 2024, doi:10.3390/life14030307_

Round 1

Reviewer 1 Report

Comments and Suggestions for Authors

The article describes relevant historical events in the history of cervical cancer and HPV.

In the introduction section, it is essential to mention that there are developing countries (Africa, Latin America, etc.) where cervical cancer is the second cause of death from cancer in women and where this neoplasia is an important public health problem.

I consider at least three figures should be included in the article. I suggest including two timelines for the topics of: From ancient promiscuity to microscopic diagnosis, and for The presence of oncogenic viruses: From "invisible poison" to viral genetics.

It is a historical article, the authors may include photographs of the leading historical figures such as George Nicolas Papanicolaou and his wife or Harald zur Hausen.

Author Response

Manuscript ID: life-2884578

Title: From Satirical poems and Invisible Poisons to Radical Surgery and

organized cervical cancer Screening - a Historical Outline of Cervical

Carcinoma and Its Relation to HPV Infection

Corresponding author: Leonard Jung; leojung95@web.de

The article describes relevant historical events in the history of cervical cancer and HPV.

We thank the reviewer for the positive assessment of our work and the helpful comments that helped us to improve our manuscript.

In the introduction section, it is essential to mention that there are developing countries (Africa, Latin America, etc.) where cervical cancer is the second cause of death from cancer in women and where this neoplasia is an important public health problem.

We added the missing information about the globally (Africa, Europe, Asia) varying incidence und consecutive implications of cervical cancer – see page 1.

I consider at least three figures should be included in the article. I suggest including two timelines for the topics of: From ancient promiscuity to microscopic diagnosis, and for The presence of oncogenic viruses: From "invisible poison" to viral genetics. It is a historical article, the authors may include photographs of the leading historical figures such as George Nicolas Papanicolaou and his wife or Harald zur Hausen.

As suggested, we added three figures to our manuscript – see Figure 1-3 – including the proposed timelines.

Reviewer 2 Report

Comments and Suggestions for Authors

This is an interesting review that is of great help in understanding the screening, diagnosis, and treatment history of cervical cancer. I would like to know if routine screening is required after HPV vaccine injection. I hope that moderate attention can be given in this regard.

Author Response

Manuscript ID: life-2884578

Title: From Satirical poems and Invisible Poisons to Radical Surgery and

organized cervical cancer Screening - a Historical Outline of Cervical

Carcinoma and Its Relation to HPV Infection

Corresponding author: Leonard Jung; leojung95@web.de

This is an interesting review that is of great help in understanding the screening, diagnosis, and treatment history of cervical cancer.

We thank the reviewer for the positive feedback and are happy to answer the comment below.

I would like to know if routine screening is required after HPV vaccine injection. I hope that moderate attention can be given in this regard.

We added the missing information on page 7: “However to date, official screening recommendations do include all women regardless of their vaccination status – since HPV vaccination is not able to provide sufficient protection against all cancerogenous HPV virus types.”

Reviewer 3 Report

Comments and Suggestions for Authors

Dear authors,

I recommend to reconsider the article after minor revision. See my remarks below.

Another comments

 Lines 112-124

In 1878, the German gynecologist Wilhelm Alexander Freund succeeded in performing an abdominal hysterectomy incluing the removal of the adnexa and part of the parametrium in a patient with cervical carcinoma [17]. Emil Reis, a student of Freund, suggested extending the technique by removing the pelvic lymph nodes [18]. This procedure of a ‘radical hysterectomy’ was summarized in 1898 by the Austrian gynecologist Ernst Wertheim and further developed by the American Joe Vincent Meigs [19]. This radical approach was accompanied by an improvement in the long-term prognosis compared to the vaginal hysterectomy introduced by Frederich Schauta, whereby latter had the advantage of a lower perioperative mortality rate in the 19th century [9]. The Wertheim-Meigs operation is still commonly used, though in order to reduce postoperative morbidity and complications experimental set ups nowadays question the need for radical surgical therapy in patients with 1A2 or 1B1 low risk cervical carcinomas with lesions smaller or equal the size of 2 cm [20].

 Comment: The history window should be slightly modified. For example:

“Surgeons operating at the time chose surgical procedures intuitively, and gradually it became clear that patients whose uterus was removed extrafascially survived longer. The work "Uterus Amputation am vaginalen Wege" from 1830 documented the description of radical vaginal hysterectomy (Ossiander, Sauter, Siebold). Vinzenz Czerny in the work "Ca colli uteri" in the Wiener medizinische Wochenschrift in 1878 used the same technique.

The first radical abdominal hysterectomy was performed in 1880 in Prague by Karel Pawlik. Both Schauta and Wertheim saw Pawlik's radical performance in Prague for the first time. Schauta modified the vaginal technique of radical hysterectomy. Thanks to the study of gonococcal infection, Wertheim assumed a similar mechanism of spread of the primary tumor into the lymphatic tissue and added systematic pelvic curative lymphadenectomy to the radical operative procedure. He described the parametria, had a special forceps made for dissecting the dorsal parametria (Wertheim Klemme). He clashed with his teacher Schauta about the meaning of lymphadenectomy, which Schauta did not perform vaginally. The so-called "Drüsenstreit", the nodule dispute, has been published in the professional European literature for decades. The dispute was ended by Wertheim's untimely death and the introduction of radiation as a method of treatment. “

 And since the title of the article is "From Satirical poems and Invisible Poisons to Radical Surgery ...", the current state of the surgical hysterectomy should be briefly mentioned. For example:

"With the advent of laparoscopy at the end of the twentieth century, there is a renaissance of the vaginal approach in combination with laparoscopic lymphadenectomy. Daniel Dargent pioneered the method and Marc Possover developed the concept to surgical perfection. Possover contributed to the introduction of the "nerve sparing technique" into the concept of TRLH (total radical laparoscopic hysterectomy). The TMMR (total mesometrial resection) method (Hoeckel, Kimmig) also belongs to the brief historical excursion. Radical hysterectomy began to be divided into types A, B, C1, C2 according to the extent of resection of parametria/paracervix. Systematic pelvic lymphadenectomy has mostly been replaced by the concept of sentinel lymph node mapping, which enables histopathological examination of SLN by ultrastaging. The increasingly popular TRLH approach has ended the LACC (the Laparoscopic Approach to Cervical Cancer) trial. Abdominal open approach is currently accepted for radical hysterectomy with SLNM or/and systematic pelvic lymphadenectomy. Management of locally advanced cervical cancer includes definitive radiotherapy with concomitant chemotherapy whenever possible. Treatment strategy should aim to avoid combining radical surgery and radiotherapy because of the high morbidity induced by the combined treatment."

 And you can add a few more sentences about immunotherapy, which is now a big trend:

"A new option in the treatment of cervical cancer is immunotherapy, which targets HPV-related viral proteins. Pembrolizumab, approved for cervical cancer patients with a PD-L1 mutation, has shown promise, with ongoing trials examining combination immunotherapies and vaccine approaches targeting HPV-related cancers. Checkpoint inhibitors combined with radiation therapy and innovative strategies like adoptive cell therapy using autologous tumor-infiltrating lymphocytes are also in development. These ongoing research efforts hold the potential to shape the future of cervical cancer management, providing hope for patients with limited treatment options. Immune checkpoint inhibitors such as pembrolizumab and nivolumab are being tested with varying response rates, and their efficacy is linked to PD-L1 expression. Additionally, Ipilimumab and other checkpoint inhibitors are being explored, necessitating further research to fully harness the potential of immunotherapy in cervical cancer treatment."

Or similarly.

Literature

Köhler C, Possover M, Tozzi R, et al. Renaissance der Operation nach Schauta. Gynäkologe. 2002;35:132- 145

Höckel M. Totale mesometriale Resektion - operative Therapie des Zervixkarzinoms auf der Grundlage einer aus der Embryonal und Fetalentwicklung abgeleiteten chirurgischen Anatomie. Geburtsh Frauenheilk. 2003;63:1146-1152

Höckel M, Wolff W, Schmidt W, et al. Nervenschonende radikale Hysterektomie I. Anatomische Grundlagen und Operationstechnik. Geburtsh Frauenheilk. 2000;60:314-319

Dastur Adi E, Tank P D. Ernst Wertheim’s radical approach to cervical cancer. J Obstet Gynecol. 2010;60(1):23-24

Ramirez PT, Frumovitz M, Pareja R, et al. Minimally Invasive versus Abdominal Radical Hysterectomy for Cervical Cancer. N Engl J Med 2018; 379(20): 1895-1904.

Querleu D, Cibula D, Abu-Rustum NR. 2017 Update on the Querleu-Morrow Classification of Radical Hysterectomy. Ann Surg Oncol. 2017 Oct;24(11):3406-3412. doi: 10.1245/s10434-017-6031-z. Epub 2017 Aug 7. PMID: 28785898; PMCID: PMC6093205.

Cibula D, Raspollini MR, Planchamp F, et al. ESGO/ESTRO/ESP Guidelines for the management of patients with cervical cancer - Update 2023. Virchows Arch. 2023 Jun;482(6):935-966. doi: 10.1007/s00428-023-03552-3. Epub 2023 May 5. PMID: 37145263; PMCID: PMC10247855.

Taha, T.; Reiss, A.; Amit, A.; Perets, R. Checkpoint Inhibitors in Gynecological Malignancies: Are we There Yet? BioDrugs 2020, 480 34, 749-762, doi: 10.1007/s40259-020-00450-x.

De Felice, F.; Marchetti, C.; Palaia, I.; Ostuni, R.; Muzii, L.; Tombolini, V.; Benedetti Panici, P. Immune check-point in cervical cancer. Crit Rev Oncol Hematol 2018, 129, 40-43, doi: 10.1016/j.critrevonc.2018.06.006

Johnson, C.A.; James, D.; Marzan, A.; Armaos, M. Cervical Cancer: An Overview of Pathophysiology and Management. Semin 509 Oncol Nurs 2019, 35, 166-174, doi: 10.1016/j.soncn.2019.02.003

Lines 215-218

Up to now over 200 genetically differentiated HPV types have been detected (of which 30 can infect the genital tract and 16 virus types are categorized as high-risk variants in the development of cervical cancer [59]), while solely 60 different HPV types were identified towards the end of the 1980s [60].

Comment: About 40 HPV genotypes are the most important clinically. The International Agency for Research on Cancer (IARC) of WHO distinguishes 3 groups of HPV. HR genotypes include HPV classified into group 1 (HPV 16, 18, 31, 33, 35, 39, 45, 51, 52, 56, 58, 59) and group 2A (HPV 68). According to the IARC, 13 HPV genotypes are currently classified into the HR HPV group, HPV genotypes 16, 18, 31, 33 and 45 are considered the most important for the development of human malignancies.

Source: Bouvard V, Baan R, Straif K et al. A review of human carcinogens--Part B: biological agents. Lancet Oncol 2009; 10(4): 321-322.

Author Response

Manuscript ID: life-2884578

Title: From Satirical poems and Invisible Poisons to Radical Surgery and

organized cervical cancer Screening - a Historical Outline of Cervical

Carcinoma and Its Relation to HPV Infection

Corresponding author: Leonard Jung; leojung95@web.de

Dear authors,

I recommend to reconsider the article after minor revision. See my remarks below.

We thank the reviewer for the thoughtful comments and all the helpful clinical information, that allowed us to improve our manuscript.

For additional details, please refer to the revised version of our manuscript. 

Lines 112-124

 Comment: The history window should be slightly modified. For example:

“Surgeons operating at the time chose surgical procedures intuitively, and gradually it became clear that patients whose uterus was removed extrafascially survived longer. The work "Uterus Amputation am vaginalen Wege" from 1830 documented the description of radical vaginal hysterectomy (Ossiander, Sauter, Siebold). Vinzenz Czerny in the work "Ca colli uteri" in the Wiener medizinische Wochenschrift in 1878 used the same technique.

The first radical abdominal hysterectomy was performed in 1880 in Prague by Karel Pawlik. Both Schauta and Wertheim saw Pawlik's radical performance in Prague for the first time. Schauta modified the vaginal technique of radical hysterectomy. Thanks to the study of gonococcal infection, Wertheim assumed a similar mechanism of spread of the primary tumor into the lymphatic tissue and added systematic pelvic curative lymphadenectomy to the radical operative procedure. He described the parametria, had a special forceps made for dissecting the dorsal parametria (Wertheim Klemme). He clashed with his teacher Schauta about the meaning of lymphadenectomy, which Schauta did not perform vaginally. The so-called "Drüsenstreit", the nodule dispute, has been published in the professional European literature for decades. The dispute was ended by Wertheim's untimely death and the introduction of radiation as a method of treatment. “

And since the title of the article is "From Satirical poems and Invisible Poisons to Radical Surgery ...", the current state of the surgical hysterectomy should be briefly mentioned. For example:

"With the advent of laparoscopy at the end of the twentieth century, there is a renaissance of the vaginal approach in combination with laparoscopic lymphadenectomy. Daniel Dargent pioneered the method and Marc Possover developed the concept to surgical perfection. Possover contributed to the introduction of the "nerve sparing technique" into the concept of TRLH (total radical laparoscopic hysterectomy). The TMMR (total mesometrial resection) method (Hoeckel, Kimmig) also belongs to the brief historical excursion. Radical hysterectomy began to be divided into types A, B, C1, C2 according to the extent of resection of parametria/paracervix. Systematic pelvic lymphadenectomy has mostly been replaced by the concept of sentinel lymph node mapping, which enables histopathological examination of SLN by ultrastaging. The increasingly popular TRLH approach has ended the LACC (the Laparoscopic Approach to Cervical Cancer) trial. Abdominal open approach is currently accepted for radical hysterectomy with SLNM or/and systematic pelvic lymphadenectomy. Management of locally advanced cervical cancer includes definitive radiotherapy with concomitant chemotherapy whenever possible. Treatment strategy should aim to avoid combining radical surgery and radiotherapy because of the high morbidity induced by the combined treatment."

Thank you very much for this detailed clinical-surgical overview – we added this topic accordingly to the revised manuscript, see page 3-4.

And you can add a few more sentences about immunotherapy, which is now a big trend:

"A new option in the treatment of cervical cancer is immunotherapy, which targets HPV-related viral proteins. Pembrolizumab, approved for cervical cancer patients with a PD-L1 mutation, has shown promise, with ongoing trials examining combination immunotherapies and vaccine approaches targeting HPV-related cancers. Checkpoint inhibitors combined with radiation therapy and innovative strategies like adoptive cell therapy using autologous tumor-infiltrating lymphocytes are also in development. These ongoing research efforts hold the potential to shape the future of cervical cancer management, providing hope for patients with limited treatment options. Immune checkpoint inhibitors such as pembrolizumab and nivolumab are being tested with varying response rates, and their efficacy is linked to PD-L1 expression. Additionally, Ipilimumab and other checkpoint inhibitors are being explored, necessitating further research to fully harness the potential of immunotherapy in cervical cancer treatment."

Thanks! We added the promising treatment options of immunotherapies to our Discussion and Future Outlook, see page 11.

Lines 215-218

Up to now over 200 genetically differentiated HPV types have been detected (of which 30 can infect the genital tract and 16 virus types are categorized as high-risk variants in the development of cervical cancer [59]), while solely 60 different HPV types were identified towards the end of the 1980s [60].

Comment: About 40 HPV genotypes are the most important clinically. The International Agency for Research on Cancer (IARC) of WHO distinguishes 3 groups of HPV. HR genotypes include HPV classified into group 1 (HPV 16, 18, 31, 33, 35, 39, 45, 51, 52, 56, 58, 59) and group 2A (HPV 68). According to the IARC, 13 HPV genotypes are currently classified into the HR HPV group, HPV genotypes 16, 18, 31, 33 and 45 are considered the most important for the development of human malignancies.

We thank the reviewer for the constructive comment. In the revised manuscript of the paper, we added extensive information about different HPV types and their malignant potential, see page 6-7 and our new Table 1.

Reviewer 4 Report

Comments and Suggestions for Authors

This narrative review focuses on 'pathological, virological, and potential future milestones' in cervical cancer screening. It is an interesting manuscript. I would suggest the authors to further discuss the role of screening in decreasing the incidence of cervical cancer. What is the incidence of CC in Europe, US, Africa....? In the last chapter the role of HPV vaccination should be described

Author Response

Manuscript ID: life-2884578

Title: From Satirical poems and Invisible Poisons to Radical Surgery and

organized cervical cancer Screening - a Historical Outline of Cervical

Carcinoma and Its Relation to HPV Infection

Corresponding author: Leonard Jung; leojung95@web.de

I would suggest the authors to further discuss the role of screening in decreasing the incidence of cervical cancer.

In accordance with the reviewer’s comment, we explain the effect of organized screening programs using the example of Northern Europe in our introduction, see page 1.

What is the incidence of CC in Europe, US, Africa....?

We added the missing information about the globally (Africa, Europe, Asia) varying incidence of cervical cancer – see page 1.

In the last chapter the role of HPV vaccination should be described.

We thank the reviewer for the constructive comment. In the revised manuscript of the paper, we did not only add more information about global vaccine types but also their role in low-income countries regarding cervical cancer, see page 7. Furthermore, we depict the necessity of ongoing screening in women who underwent HPV vaccination (page 7).

Moreover, we added information about different HPV types and their malignant potential, see page 6-7 and our new Table 1.
